# Smartphone-Based Analysis for Early Detection of Aging Impact on Gait and Stair Negotiation: A Cross-Sectional Study

**DOI:** 10.3390/s25072310

**Published:** 2025-04-05

**Authors:** Roee Hayek, Rebecca T. Brown, Itai Gutman, Guy Baranes, Shmuel Springer

**Affiliations:** 1The Neuromuscular & Human Performance Laboratory, Department of Physical Therapy, Faculty of Health Sciences, Ariel University, Ariel 4070000, Israel; roeeh@ariel.ac.il (R.H.); itaigu@ariel.ac.il (I.G.); guybaranes101@gmail.com (G.B.); 2Division of Geriatric Medicine, Perelman School of Medicine, University of Pennsylvania, Philadelphia, PA 19104, USA; rebecca.brown@pennmedicine.upenn.edu; 3Geriatrics and Extended Care Program, Corporal Michael J. Crescenz VA Medical Center, Philadelphia, PA 19104, USA; 4Center for Health Equity Research and Promotion, Corporal Michael J. Crescenz VA Medical Center, Philadelphia, PA 19104, USA; 5Leonard Davis Institute of Health Economics, University of Pennsylvania, Philadelphia, PA 19104, USA; 6Research Associate Canadian Center for Activity and Ageing, University of Western Ontario, London, ON N6A 3K7, Canada

**Keywords:** middle-age, smartphone-accelerometry, gait, stairs, aging, mobility

## Abstract

Aging is associated with gradual mobility decline, often undetected until it affects daily life. This study investigates the potential of smartphone-based accelerometry to detect early age-related changes in gait and stair performance in middle-aged adults. Eighty-eight healthy participants were divided into four age groups: young (20–35 years), early middle-aged (45–54 years), late middle-aged (55–65 years), and older adults (65–80 years). They completed single-task, cognitive, and physical dual-task gait assessments and stair negotiation tests. While single-task walking did not reveal early changes, cognitive dual-task cost (DTC) of stride time variability deteriorated in late middle age. A strong indicator of early mobility changes was movement similarity, measured using dynamic time warping (DTW), which declined from early middle age for both cognitive DTC and stair negotiation. These findings highlight the potential of smartphone-based assessments, particularly movement similarity, to detect subtle mobility changes in midlife, allowing for targeted interventions to promote healthy aging.

## 1. Introduction

Aging is frequently associated with changes in mobility, often beginning with subtle, preclinical limitations that may go unnoticed by those affected. Over time, these changes can progress into significant locomotion disabilities, severely impacting daily activities and overall quality of life [1,2]. Maintaining adequate mobility is critical for promoting healthy aging [3]. Consequently, it is vital to identify which aspects of mobility deteriorate most rapidly and at what stages of life.

Aging is a nonlinear process, with significant biologic, metabolic, and physiologic changes occurring in early and late middle age [4]. Although most mobility impairments are associated with old age, they can start as early as middle age. Up to 20% of the middle-aged population report mobility difficulties despite having only mild and general health problems [5,6]. Studies in healthy middle-aged adults have shown a deterioration in some aspects of dynamic balance abilities, with differences between early and late middle-aged adults [7,8]. Evidence from population-based cohort studies indicates a marked increase in fall prevalence within this age group [9,10]. Consequently, recent research underscores the importance of assessing mobility in midlife [11,12].

Analysis of spatiotemporal characteristics of mobility provides comprehensive insight into the biomechanical and neuromuscular changes associated with aging [13,14,15]. Walking speed is widely recognized in aging research as a ‘sixth vital sign’ that correlates strongly with overall health [16,17]. While comfortable walking speed may not reveal early subtle changes in mobility, as it declines at age 65 and becomes more pronounced after age 70 [13,18,19], dual-task gait velocity may begin to decline around age 50 [20,21]. Variability and similarity in spatiotemporal mobility characteristics, quantified respectively by metrics such as stride time variability and dynamic time warping (DTW), a method of comparing time series data [22,23], are also important measures associated with gait stability and age-related locomotion decline [13,24]. Older adults frequently exhibit increased gait variability compared to young adults [13,14,24,25], whereas gait variability in middle age remains largely unchanged during single-task walking [13,14,24] and deteriorates significantly in dual-task scenarios [21,23]. Yet, most research has focused on comparing older and younger adults. Further investigation into mobility performance across the adult lifespan, including midlife, is essential for identifying early markers of functional decline, as highlighted in a systematic review by Herssens et al. [13].

Dual-task cost (*DTC*), defined as the difference in mobility performance between single and dual tasks, is an important measure for evaluating locomotion ability [26]. Zhou et al. [21] found that *DTC* increases with age, particularly in cognitive tasks such as serial subtraction. However, this task often lacks ecological validity, emphasizing the need to study more natural scenarios, such as texting while walking [20,27]. Furthermore, everyday locomotion frequently involves physical dual tasks, such as carrying loads, which may also affect performance, particularly in older adults [28,29]. Several studies show that carrying 10% or more of body weight impairs locomotion and postural control even in young adults [29,30]. Yet, the effects of external loading on mobility in different age groups have not been adequately studied.

Body-worn sensors, particularly accelerometers, provide accurate, low-cost measurements of mobility in real-world settings with high ecological validity and can support early detection of mobility impairment [31,32]. To date, accelerometers are integrated into all smartphones, providing a more accessible and affordable alternative to specialized devices [11,33]. Studies confirm that smartphones can accurately measure spatiotemporal gait parameters, including velocity, stride variability, and muscle power, to effectively detect age-related changes [11,34,35]. In addition, most research on the effects of aging on mobility focuses on walking, although the use of new technologies and mobility metrics may reveal changes in stair negotiation that requires greater balance, coordination, and strength [36,37]. Similar to gait, accelerometry can detect early mobility changes in stair performance by assessing movement patterns and muscle power while overcoming the ceiling effects of time-based assessments [38,39]. However, research in middle age is still limited [13].

This study aimed to examine whether smartphone-based analysis can detect age-related mobility changes during walking and stair negotiation. We hypothesized that stride time variability and *DTW* during walking with a physical or cognitive dual task, as well as muscle power and vertical axis similarity (*DTW*) during stair negotiation, would be negatively changed starting in late middle age, while deterioration in gait speed and stair ascent and descent times would only be observed in older adults.

## 2. Methods

### 2.1. Participants

Eighty-eight participants were recruited for the study, divided evenly into four age groups (n = 22 per group): young adults (20–35 years), early middle-aged adults (45–54 years), late middle-aged adults (55–65 years), and older adults (65–80 years). Sample size was based on a power analysis using G*Power 3.1.9 [40], for a one-way ANOVA comparison between four age groups (power = 0.8, alpha = 0.05, large effect size, f = 0.4). An effect size of f = 0.4 was chosen based on the exploratory nature of the study and prior evidence suggesting substantial group-level differences in mobility parameters between young and middle-aged adults [11,41,42]. This analysis indicated that a minimum sample size of 76 participants would be required to detect differences between groups, and previous research suggested that at least 20 participants per group are needed [11,41,42]. Two participants (one from the early middle-aged and one from the older adult groups) were excluded from the final analysis due to measurement issues. Participants were university students, university employees, and local residents screened via telephone. Individuals were included if they were between the ages of 20 and 35 or 45 and 80 years, lived independently in the community, were able to walk outdoors, and performed moderate to vigorous physical activity. Those with neurologic, orthopedic, vestibular, or significant visual impairments (e.g., post-stroke, Parkinson’s disease, cancer, recent fracture or orthopedic surgery, age-related macular degeneration, glaucoma, or diabetic retinopathy) that could affect mobility were excluded based on a pre-enrollment telephone interview in which medical history was reviewed. The study was approved by the Ariel University Ethics Committee (AU-HEA-SS−20230806), and all participants provided written informed consent.

### 2.2. Procedure

Each participant completed a single 30–45-min test session. Participants were instructed to dress comfortably and wear flat shoes. Anthropometric data were initially collected, and then mobility assessments were performed in a randomized order. Gait was evaluated with and without dual cognitive and physical tasks, and stair negotiation was assessed during ascent and descent. For the measurement, a smartphone (Galaxy A73; Android 13; 163 × 76.1 × 7.6 mm; 181 g) was attached with an elastic belt to the participants’ lumbar spine near the body’s center of mass, a validated and reliable position for data collection (as shown in Figure 1a) [34]. Acceleration data were recorded using the phyphox app (100 Hz, RWTH, Aachen, Germany), which was remotely controlled via Wi-Fi [43,44]. Due to temporary internet interruptions, a limited number of measurements were incomplete, resulting in missing data in some cases, which are detailed in the Results section.

Gait was assessed while participants walked along a circular corridor (as shown in Figure 1b) at a comfortable speed for two minutes, with the first five meters used for acceleration and not recorded. The cognitive dual task required participants to provide rapid text responses to questions created using a chatbot generator (https://landbot.io/, accessed on 28 January 2024). The task consisted of sequential questions on personal and general topics (e.g., place of residence, hobbies, or sports), as asked in previous studies where texting was used as a dual task while walking [20,45]. Answers were kept short, with participants typing “Not relevant” for irrelevant questions. Respondents’ answers were not recorded. Participants used their own smartphones for texting, without specific instructions on how to hold them. In the physical dual-task condition, participants walked while carrying in their dominant hand a cloth shopping bag with free weights equivalent to 10% of their body weight.

In the measurement of stair negotiation, participants were asked to climb up and down a staircase with thirteen steps (16 cm high, 30 cm deep, and 155 cm wide, as shown in Figure 1c) at a self-selected pace, avoiding handrails, skipping steps, or double-foot placements. The instructions were: “Go up or down the stairs at your comfortable pace, just as you would in everyday life. At the bottom of the stairs, stand with your back to the stairs for three seconds to ensure the end of the recording”.

### 2.3. Gait and Stairs Negotiation Measures and Data Processing

Gait measures included: gait speed (m/sec), stride time variability (%), calculated as standard deviation of stride time relative to average stride duration [46]; and *DTW* of the acceleration signal in the anteroposterior (AP) and mediolateral (ML) axes [47]. Stair negotiation outcomes included: total ascent and descent time (sec) and muscle power (watts), calculated as subject body mass (kg) multiplied by vertical acceleration (m/s²) multiplied by vertical velocity (m/s) [48]. To normalize muscle power, it was divided by subject’s body mass (watts/kg) [11]. *DTW* was used to assess the consistency of movement patterns [49,50].

Signal processing and parameter extraction for the gait and stair negotiation assessments were performed using MATLAB (MathWorks, Inc.; version 9.12, Natick, MA, USA). Spatiotemporal gait characteristics were calculated as previously described [34]. The accelerometer data were resampled to 100 Hz and filtered with a 4th-order Butterworth low-pass filter (20 Hz cutoff). Additionally, the AP acceleration signal was filtered with a 2 Hz cutoff, and heel strikes were identified from the positive AP peaks using the MATLAB function ‘findpeaks’. The processing of stair negotiation data followed a similar approach to gait analysis but focused on step analysis instead of stride analysis and on vertical acceleration rather than AP acceleration. Local maxima and minima were used to identify contacts for each step. The step cycle was defined as the interval between the initial contact of the leading foot on the step and the subsequent contact of the trailing foot [51].

To assess the effects of the cognitive and physical tasks on gait performance compared to a single-task condition, the *DTC* of the cognitive (*DTC* cognitive) and physical (*DTC* physical) tasks, were calculated for each gait measure, using the following formula [21]:$$DTC=\left(\frac{Performance\;under\;single\;task-performance\;under\;dual\;task}{performance\;under\;single\;task}\right)\times 100$$

A positive *DTC* of gait speed indicates slower walking under dual-task conditions, reflecting a negative impact of the additional task. In contrast, a negative *DTC* of stride time variability or *DTW* indicates increased variability or decreased similarity, suggesting a negative influence of the dual task.

### 2.4. Statistical Analysis

Normal distribution was assessed using the Shapiro–Wilk test and histograms. Quantitative variables were summarized for descriptive statistics using mean and standard deviation (SD) or median and interquartile range (IQR), as appropriate. Categorical variables are presented as frequencies and percentages.

Background variables (e.g., gender, anthropometrics) were compared between groups using one-way analysis of variance (ANOVA) to examine possible covariates. To identify age-related mobility changes, gait measures from the single-task condition, the *DTC* of cognitive and physical tasks, and stair negotiation performance were compared across groups using the Kruskal–Wallis test, as the assumptions of normality and homogeneity of variances were not met. Pairwise comparisons were conducted using the Dunn–Bonferroni approach to determine the age group at which mobility changes become evident across different measures. To quantify the magnitude of observed differences, effect sizes for the pairwise comparisons were calculated using Cohen’s r for non-parametric data, where 0.1, 0.3, and 0.5 indicate small, medium, and large effects, respectively [52]. Outliers in the measured values (2.5 times the IQR) were excluded to ensure robustness. In addition, to further explore aspects of movement control during aging, we also examined two key relationships using Spearman’s correlation: (1) cognitive demand—the relationship between movement similarity when climbing stairs (measured by *DTW*) and cognitive-motor interference when walking with cognitive dual task (cognitive *DTC* of *DTW*), i.e., a reduction in motor performance due to the additional demand of a simultaneous cognitive task [53], and (2) muscular demand—the association between movement similarity (*DTW*) and muscle power during stair climbing. Statistical analysis was performed using IBM SPSS Statistics, version 27.0 (IBM Corp, Armonk, NY, USA). Significance was set at *p* < 0.05.

## 3. Results

### 3.1. Baseline Characteristics

The baseline Characteristics of the participants are shown in Table 1. There were no differences between the groups, except for age.

### 3.2. Gait and Stair Negotiation

The gait measures of the single-task condition and the *DTC* of the cognitive and physical tasks, as well as the number of missing values in the gait measures, are shown in Table 2, and the stair negotiation measures, including the number of missing values, are shown in Table 3. The results of each test are explained in more detail below.

### 3.3. Gait Velocity

Gait speed under the single-task condition differed significantly between groups (*p* = 0.004). Post-hoc analysis revealed that only young adults walked significantly faster than older adults (*p* < 0.001, Cohen’s r = 0.56). The effect of the cognitive task on gait speed (i.e., cognitive *DTC* of gait speed) was similar between age groups (*p* = 0.143). However, the physical *DTC* of gait speed showed age-related differences (*p* < 0.001). Older adults had significantly different physical *DTC* than young adults (*p* < 0.001, Cohen’s r = 0.65) and early middle-aged adults (*p* < 0.001, Cohen’s r = 0.74). Notably, they were the only group with a negative physical *DTC* (−3.56%, IQR: −8.56 to −1.14). Figure 2a illustrates the distribution of the cognitive *DTC* of gait velocity across all age groups.

### 3.4. Gait Variability

Stride time variability during single-task walking differed significantly between age groups (*p* < 0.001). Late middle-aged adults had the lowest variability compared to all three other groups, including young adults (*p* = 0.018, Cohen’s r = 0.45), early middle-aged adults (*p* = 0.014, Cohen’s r = 0.46), and older adults (*p* > 0.001, Cohen’s r = 0.75). The effect of the cognitive task on stride time variability differed significantly between age groups (*p* = 0.009), with late middle-aged adults having significantly higher cognitive *DTC* of stride time variability than young adults (*p* = 0.006, Cohen’s r = 0.50), indicating a greater increase in variability during the cognitive task compared to the single-task condition. In contrast, the physical *DTC* of stride time variability did not differ between age groups (*p* = 0.060). Figure 2b presents the distribution of the cognitive *DTC* of stride time variability across all age groups.

### 3.5. Gait Similarity

No between-group differences in *DTW* were observed during single-task walking in the AP or ML directions (*p* > 0.05). The effect of the cognitive task on *DTW* showed significant differences between groups only in the AP direction (*p* = 0.002). The young adults showed significantly lower cognitive *DTC* of *DTW-AP* compared to all other three age groups, early middle-age (*p* = 0.035, Cohen’s r = 0.42), late middle-age (*p* = 0.007, Cohen’s r = 0.50) and older adults (*p* = 0.006, Cohen’s r = 0.51). No other significant differences were found between the groups. Figure 2c illustrates the distribution of the cognitive *DTC* of *DTW* across all age groups.

### 3.6. Stairs Ascend and Descend Time

The analysis of stair negotiation times showed no statistically significant differences between the four age groups, neither for the total stair ascent time (*p* = 0.445) nor for the total stair descent time (*p* = 0.669).

### 3.7. Muscle Power During Stairs Negotiation

No significant differences were found between the age groups in muscle power during the stair ascent (*p* = 0.294). In contrast, muscle power during stair descent differed significantly between age groups (*p* < 0.001), with older adults showing significantly lower muscle power than early middle-aged adults (*p* = 0.007, Cohen’s r = 0.51) and young adults (*p* = 0.001, Cohen’s r = 0.57).

### 3.8. Stairs-Negotiation Similarity

Significant differences in *DTW* were observed between age groups during both stair ascent and stair descent (*p* < 0.001, in both conditions). During stair ascent, young adults showed significantly better similarity (e.g., lower *DTW*) compared to all other three age groups, early middle-age (*p* < 0.001, Cohen’s r = 1.09), late middle-age (<0.001, Cohen’s r = 0.87), and older adults (<0.001, Cohen’s r = 0.63). In addition, early middle-aged adults had significantly better similarity than older adults (*p* = 0.02, Cohen’s r = 0.46). During stair descent, young adults also showed significantly better similarity compared to all other three age groups, including early middle-age (*p* < 0.001, Cohen’s r = 0.86), late middle-age (<0.001, Cohen’s r = 0.92), and older adults (<0.001, Cohen’s r = 0.77). Figure 3 illustrates the age-related effects on stair ascent and descent, presenting the distributions of duration and movement similarity across age groups.

### 3.9. The Relationship Between Movement Similarity of Stair Climbing to the Cognitive DTC of Walking and Muscle Power

The correlation analyses conducted to further investigate aspects of movement control during aging revealed different patterns across age groups. Similarity of movement during stair climbing (*DTW*) was positively correlated with cognitive-motor interference during walking (cognitive *DTC* of *DTW*) in early middle-aged (r = 0.585, *p* = 0.007) and older adults (r = 0.563, *p* = 0.012). No significant correlations were found in young adults or late middle-aged adults. While muscle power did not correlate significantly with *DTW* during stair climbing in any group, a trend was observed in young adults in which muscle power correlated negatively with the similarity of movement during stair climbing (r = −0.403, *p* = 0.07).

## 4. Discussion

This study highlights the utility of smartphone accelerometry for early detection of subtle changes in mobility associated with aging. It suggests that analyzing cognitive dual-task costs of stride time variability and similarity metrics during gait, and movement similarity when negotiating stairs, can serve as indicators of age-related mobility decline, with notable changes beginning in middle age. Our findings indicate that gait tests without dual-task conditions cannot effectively detect early mobility changes, whereas tests including dual-task conditions, particularly those involving cognitive-motor interference and performance in stair negotiation, can reveal these subtle changes. These insights support the real-time application of smartphone-based mobility assessments or accelerometry-based wearables for early detection and targeted interventions.

Our findings show that under the single-task condition, gait speed, variability, and similarity remained relatively stable in middle-aged adults. As expected, only older adults exhibited slower gait speed compared to young adults. However, it should be noted that although the gait speed of both middle-aged groups did not differ from that of young adults, it also did not differ from that of older adults, which may suggest a subtle decline. Our findings are aligned with previous studies [13,18,19,54] reporting a gradual decline in walking speed with aging, with significant changes over 70 years. Even under cognitive dual-task conditions, walking speed did not reveal age-related differences, consistent with Hennah & Doumas [55], and Yogev-Seligmann et al. [56] who found that cognitive-motor interference of walking speed does not adequately capture age-related decline. This supports the notion that while walking speed is a key biomarker of aging, it may not be sensitive enough to capture preclinical mobility changes in midlife [54,57].

Interestingly, during the single-task condition, late middle-aged adults showed the lowest stride time variability compared to all groups, while older adults showed no deterioration compared to young adults. Nevertheless, stride time variability was within the normative range of 1 to 2% in both the young and middle-aged groups as reported previously [58], whereas older adults had a median value of 2.6%, which is the upper bound of non-pathologic gait [59,60]. Furthermore, the comparison between late middle-aged adults and older adults was the only between groups comparison of stride time variability that showed a large effect. In contrast to the single-task condition, the cognitive task had a greater impact on late middle-aged adults’ stride time variability, as indicated by the difference in *DTC* compared to young adults. Zhou et al. [21] also reported that a cognitive task had a significant impact from the age of 50 onwards, leading to greater dual-task costs in stride time variability. Similarly, Morrison et al. [61] emphasized that gait variability under cognitive dual-task condition may serve as a sensitive marker for detecting early mobility decline in middle-aged adults. It is possible that older adults did not show greater dual-task cost than young adults because they already had high variability in the single-task condition, which may have attenuated the effects of the cognitive task. A similar pattern was observed by Naaman et al. [42] in which middle-aged subjects were more affected by reduced sensory input due to reduced lighting during walking compared to young and older adults.

A strong indicator of early mobility changes in our study was the *DTC* of gait similarity in the AP direction which demonstrates a decline from early middle age. In contrast to previous studies [62,63], we could not detect an aging effect on gait similarity in the ML direction. However, these previous studies analyzed the root mean square (RMS) or coefficient of variation (CoV) of step width rather than similarity. Furthermore, texting while walking has been shown to reduce ML acceleration even in young adults, likely due to a compensatory mechanism to improve stability [64], which may further explain the lack of differences between groups in ML direction. Future studies conducted in dynamic, real-world settings are needed to extend these findings.

While we hypothesized that movement similarity during stair negotiation would decline from late middle age, our findings suggest that deterioration begins in early middle age for both ascent and descent. Our findings suggest that movement similarity measured by *DTW* may provide insight into early deterioration of mobility in middle-age. Although variability is a more commonly used metric, the measure of movement similarity can offer distinct advantages in diagnosis, as it is able to detect subtle structural changes in movement patterns [65]. Variability, which calculates the ratio of the standard deviation to the mean, provides a scalar assessment that focuses on the magnitude of variation. However, this linear approach does not take into account the complex structural patterns of movements and their dynamic stability [66,67]. In contrast, *DTW* is a non-linear method that compares entire movement sequences over time [65], enhancing variation detection and making it a more comprehensive tool for assessing age-related mobility changes [22]. In addition, while measures of variability often require extensive step data, similarity-based approaches such as *DTW* can provide meaningful insights even with fewer steps [68,69], which is particularly relevant in stair negotiation where step counts are inherently limited. For these reasons, and based on our results, we recommend the use of *DTW* metrics to detect early mobility changes. However, as this is the first data from a cross-sectional study, our results should be interpreted cautiously and further research is needed to confirm our findings.

Contrary to our hypothesis, the task of carrying loads showed almost no age-related effects on gait. It is possible that similar to Bampouras & Dewhurst findings with 3 kg [70], our 10% of body weight load may have been insufficient to produce observable changes in mobility, possibly due to the relatively high functioning of our sample, which included active and independent adults who may not represent the broader population. The only notable effect was a negative *DTC* of gait speed in older adults, suggesting a possible increase in their walking speed. This may indicate that the task was too challenging for the older group, so they unconsciously made an effort to complete the task faster or exerted more effort to cope with the additional load [71]. Further research is needed in more diverse population.

As hypothesized, the duration of stair ascent and descent did not demonstrate early changes in mobility, and, contrary to our hypothesis, it did not change even in older adults. Previous studies testing ascent times of stairs as quickly as possible showed no differences between middle-aged to young adults, while ascent times of older adults increased [38,39]. This discrepancy may be related to the different number of stairs tested in each study, as well as the instruction to ascend at maximum speed as opposed to a self-selected speed.

Another key outcome of aging measured in our study was lower limb muscle power, which is more sensitive for detecting early age-related changes than muscle force or mass [11]. Contrary to our hypothesis, late middle-aged adults showed no deterioration in muscle power when ascending or descending stairs, whereas older adults had reduced muscle power when descending stairs compared to young and early middle-aged adults, but not compared with late middle-aged adults, indicating a possible subtle trend. This may be explained by the greater demands on eccentric contraction during descent, as older adults have been reported to have lower peak acceleration in this task [36].

Detecting early age-related decline in muscle power may depend on the intensity of the task. Studies investigating stair ascent at maximal speed have shown a decline in muscle power from middle age [38,72], while our results and those of Psaltos et al. [73] who also measured stair ascent at natural speed, have found no age-related effect. This may suggest that only high-intensity, sufficiently demanding tasks can effectively detect an early age-related decline in muscle power. Similarly, Hayek et al. [11] reported a decline in muscle power in middle-aged adults only when getting up from a cushioned sofa, which is considered a demanding task, but not when getting up from a regular chair at standard height.

We found a significant positive correlation between the similarity of movement during stair climbing with the level of cognitive-motor interference during walking in early middle-aged and older adults. This may indicate a relationship between the ability to cope with cognitive load and motor performance during early aging, which has already been established in studies with older adults [74,75]. In young adults, we observed a trend towards a negative correlation between muscle power and movement similarity of during stair climbing. This may suggest that greater muscle may contribute to more consistent movement of young adults, aligning with research highlighting its key role in functional performance in this age group [39,72]. While there was no correlation in the late middle-aged group, our findings warrant further investigation.

This study has several limitations. First, the sample size was designed to detect large effects, yet a larger sample may be needed to detect more subtle age-related changes in mobility, particularly in midlife, where individual differences in human performance are substantial. Second, the exclusion of participants with certain medical conditions may limit the generalizability of the findings to broader populations. Moreover, as the study included only functionally independent participants, the results could be generalized to individuals with lower functional status. In addition, the cross-sectional design limits the ability to examine longitudinal changes or establish causality in age-related mobility declines. Future research with larger, more diverse samples and longitudinal designs is recommended to confirm and extend these findings. Despite these limitations, the results clearly demonstrate the potential of smartphone accelerometry to detect subtle early mobility changes.

## 5. Conclusions

We have demonstrated the potential usefulness of smartphone-based accelerometry in detecting early mobility changes using metrics such as movement variability and similarity and the cognitive dual-task paradigm. Further research is warranted to refine our findings with larger and more diverse populations in real-life scenarios and in a longitudinal design. Our methods, which demonstrate the ability to detect early subtle mobility changes, could serve as a screening tool to address these changes and potentially improve quality of life in later years.

## Figures and Tables

**Figure 1 sensors-25-02310-f001:**
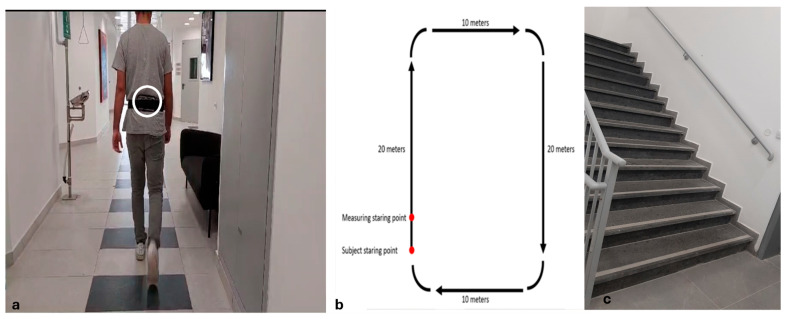
The smartphone location and the measurement environments. (**a**) Participant walking with a smartphone attached to the lower back via an elastic belt for motion data collection. (**b**) Schematic of the circular corridor used for gait assessment under single and dual-task conditions. (**c**) Staircase (13 steps; 16 cm height, 30 cm depth, 155 cm width) used for ascent and descent evaluation at a self-selected pace.

**Figure 2 sensors-25-02310-f002:**
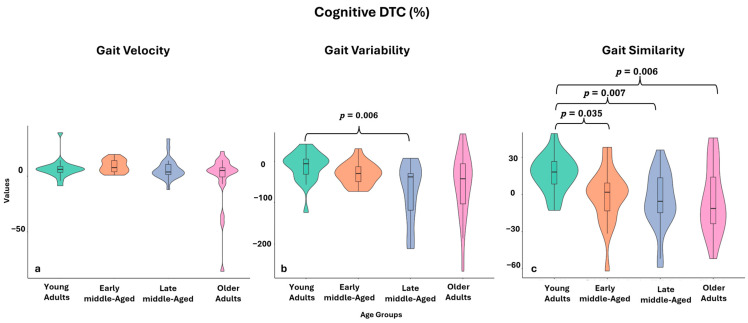
Distribution of cognitive DTC (%) for gait velocity, variability, and similarity across age groups. Notes: DTC = dual-task cost. (**a**) cognitive DTC of gait velocity (%); (**b**) cognitive DTC of stride time variability (%); (**c**) cognitive DTC of DTW (%). The violin diagram shows the distribution of the groups: green for young adults, orange for early middle-aged, blue for late middle-aged, and purple for older adults. The box within the violin represents the interquartile range (IQR), with the center line indicating the median.

**Figure 3 sensors-25-02310-f003:**
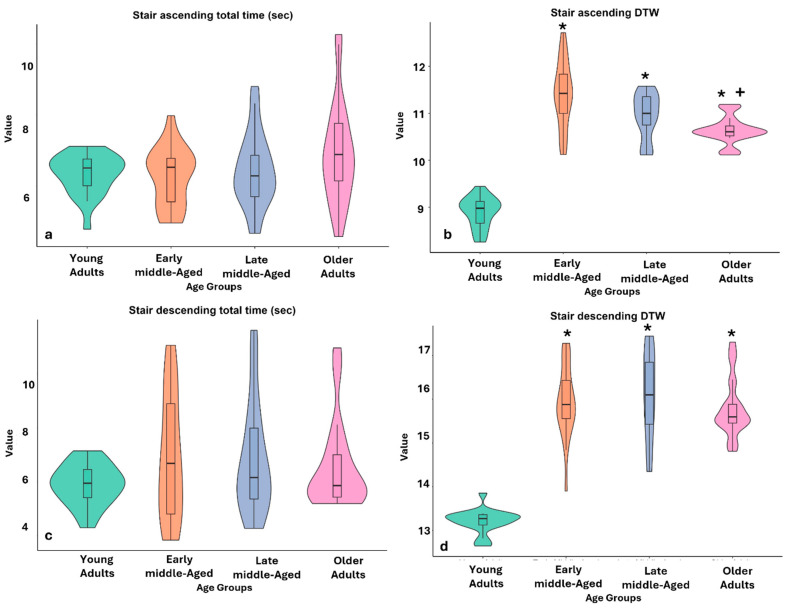
Age-related effects on stair ascent and descent: distributions of duration and movement similarity. Notes: DTW = Dynamic time warping. (**a**) Stair ascending total time (s); (**b**) DTW during stair ascent; (**c**) stair descending total time (in seconds); (**d**) DTW values during stair descent. The violin diagram shows the distribution of the groups: green for young adults, orange for early middle-aged, blue for late middle-aged, and purple for older adults. The box within the violin represents the interquartile range (IQR), with the center line indicating the median. Asterisks (*) indicate *p* < 0.001 compared to the young adult group, and plus signs (+) indicate *p* < 0.001 compared to the middle-aged group.

**Table 1 sensors-25-02310-t001:** Participants’ Baseline Characteristics.

	Young Adults (n = 22)	Early Middle-Aged Adults (n = 21)	Late Middle-Aged Adults (n = 22)	Older Adults (n = 21)	*p*-Value
Age (years)	24.7 ± 2.9	48.5 ± 2.8	59.8 ± 3.0	71.9 ± 4.6	<0.001
Female, n (%)	12 (54%)	12 (51%)	10 (45%)	11 (52%)	0.884
Height (m)	1.69 ± 0.1	1.65 ± 0.1	1.68 ± 0.1	1.64 ± 0.1	0.372
Weight (kg)	70.4 ± 14.1	72.7 ± 14.0	77.7 ± 16.3	69.2 ± 12.2	0.528
Body mass index (kg/m^2^)	25.4 ± 2.7	26.5 ± 3.2	26.9 ± 3.6	25.7 ± 4.4	0.129

Note: SD = standard deviation.

**Table 2 sensors-25-02310-t002:** Gait Variables.

Variable		Young Adults (Group #1, n = 22)	Early Middle-Aged (Group #2, n = 21)	Late Middle-Aged (Group #3, n = 22)	Older Adults (Group #4, n = 21)	Kruskal-Wallis H, *p*-Value	Pairwise Comparisons Compared Group, *p*, ES
Gait Velocity	Single task (m/s)	1.17 (1.14–1.18)	1.12 (1.10–1.17)	1.13 (1.09–1.15)	1.04 (0.91–1.16)	13.589, 0.004	1 vs. 4: *p* = 0.002, ES = 0.56
	DTC Cognitive (%)	6.08 (2.83–9.11)	7.65 (3.63–14.41)	3.75 (1.12–11.47)	5.06 (−3.47–9.96)	5.422, 0.143	None
	DTC Physical (%)	1.90 (−0.00–3.24)	3.05 (0.55–5.40) *	0.25 (−1.20–1.51) +	−3.56 (−8.56 –- 1.14)	27.332, <0.001	1 vs. 4: *p* < 0.001, ES = 0.65; 2 vs. 4: *p* < 0.001, ES = 0.74
Stride Time Variability	Single task (%)	2.00 (1.69–2.46)	1.98 (1.74–2.45)	1.55 (1.37–1.92)	2.65 (2.09–4.20)	25.454, <0.001	3 vs. 1: *p* = 0.018, ES = 0.45; 3 vs. 2: *p* = 0.014, ES = 0.46; 3 vs. 4: *p* < 0.001, ES = 0.75
	DTC Cognitive (%)	−5.93 (−34.17–6.55)	−29.45 (−49.65–−9.58) *	−37.73 (−120.53–−22.78) *	−42.24 (−111.00–−2.37) *	11.575, 0.009	1 vs. 3: *p* = 0.006, ES = 0.50
	DTC Physical (%)	17.38 (3.84–30.22)	10.55 (−0.12–21.19) *	−8.33 (−19.52–15.87) *	13.10 (−23.61–45.20)	7.412, 0.060	None
DTW AP	Single task	9.45 (9.15–11.19)	8.66 (7.59–10.12)	8.46 (7.73–10.07)	9.83 (7.98–12.84) *	7.967, 0.047	None
	DTC Cognitive (%)	16.22 (7.96–27.10)	1.81 (−16.26–10.53)	−5.53 (−15.84–15.564)	−11.30 (−26.22–23.28) +	14.936, 0.002	1 vs. 2: *p* = 0.035, ES = 0.42; 1 vs. 3: *p* = 0.007, ES = 0.50; 1 vs. 4: *p* = 0.006, ES = 0.51
	DTC Physical (%)	−4.63 (−18.06–10.63)	−0.40 (−12.70–4.71)	2.37 (−9.53–10.73) *	−8.19 (−35.94 –12.37) +	1.342, 0.724	None
DTW ML	Single task	12.19 (10.32–14.74)	11.64 (9.51–13.48)	11.56 (9.28–13.15) *	13.08 (11.57–15.23) +	5.560, 0.135	None
	DTC Cognitive (%)	12.59 (−6.24–24.39)	−5.98 (−20.51–11.43)	−5.63 (−23.06–9.67) *	2.18 (−0.98–19.25) +	7.145, 0.067	None
	DTC Physical (%)	−8.35 (−22.44–5.23)	−7.39 (−19.33–9.64)	−3.76 (−19.69–4.36) *	9.57 (−19.43–17.12) +	3.730, 0.292	None

Notes: DTW = Dynamic time warping; DTC = dual-task cost; ML = mediolateral; AP = anteroposterior; ES = effect size, analyzed with Cohen’s r. Values are medians and interquartile ranges unless otherwise indicated. A Kruskal–Wallis test was used for between age groups comparison, and *p* values in pairwise comparisons were adjusted using the Dunn–Bonferroni approach. * One missing value; + Two missing values.

**Table 3 sensors-25-02310-t003:** Stair Negotiation Variables.

Condition	Variable	Young Adults (Group #1, n = 22)	Early Middle-Aged (Group #2, n = 21)	Late Middle-Aged (Group #3, n = 22)	Older Adults (Group #4, n = 21)	Kruskal-Wallis H, *p*-Value	Pairwise Comparisons
Ascend	Total time (s)	6.50 (5.94–6.77)	6.52 (5.52–6.80)	6.48 (5.69–7.00)	6.98 (5.90–7.83) *	2.260, 0.445	None
	Muscle Power normalized to body weight (watts/kg)	3.32 (3.17–3.63) *	3.47 (3.10–3.85)	3.23 (2.78–3.79)	3.65 (3.13–4.05)	3.713, 0.294	None
	DTW	8.98 (8.65–9.13)	11.42 (10.97–11.92) *	10.99 (10.63–11.36)	10.61 (10.51–10.76)	57.126, <0.001	1 vs. 2: *p* < 0.001, ES = 1.09; 1 vs. 3: *p* < 0.001, ES = 0.87; 1 vs. 4: *p* < 0.001, ES = 0.63; 2 vs. 4: *p* < 0.001, ES = 0.46
Descend	Total time (s)	5.94 (5.37–6.50)	6.08 (4.74–9.07)	6.15 (5.21–8.06)	6.76 (5.37–7.15) *	1.558, 0.669	None
	Muscle Power normalized to body weight (watts/kg)	2.08 (1.93–2.30)	2.02 (1.80–2.35)	1.81 (1.45–2.24)	1.65 (1.53–1.91) +	17.240, <0.001	1 vs. 4: *p* < 0.001, ES = 0.57; 2 vs. 4: *p* = 0.007, ES = 0.51
	DTW	13.27 (13.11–13.36) +	15.88 (15.48–16.44) *	16.10 (15.40–16.93)	15.59 (15.45–16.11)	44.342, <0.001	1 vs. 2: *p* < 0.001, ES = 0.86; 1 vs. 3: *p* < 0.001, ES = 0.92; 1 vs. 4: *p* < 0.001, ES = 0.77

Notes: DTW = Dynamic time warping; DTC = dual-task cost; ES = effect size, analyzed with Cohen’s r. Superscript was used to show pairwise comparisons using post hoc analysis between age groups using the Kruskal–Wallis test with median (interquartile range). A Kruskal–Wallis test was used for between age groups comparison, and *p*-values were adjusted using the Dunn–Bonferroni approach. * One missing value; + Two missing values.

## Data Availability

The data supporting the findings of this study are available from the corresponding author upon reasonable request.

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
