# Peer review of "Smartphone-Based Analysis for Early Detection of Aging Impact on Gait and Stair Negotiation: A Cross-Sectional Study"

_sensors, 2025, doi:10.3390/s25072310_

Round 1
Reviewer 1 Report
Comments and Suggestions for Authors
This paper investigates the potential of smartphone-based accelerometry to detect early age-related mobility changes, particularly in gait and stair negotiation, with a specific focus on middle-aged adults. The study is motivated by the need for early detection of mobility decline, which often goes unnoticed until it significantly impacts daily life. The authors hypothesize that certain gait parameters, such as stride time variability and movement similarity, would reveal subtle changes in midlife, offering a basis for early intervention. The English in the paper is clear, professional, and generally well-written. The study is a well-executed and timely contribution to the growing field of wearable-based mobility assessments. Its key strength lies in demonstrating that subtle changes in movement similarity and cognitive dual-task costs can be detected in midlife, paving the way for early, smartphone-based screening tools. However, to maximize clinical impact, future research should address longitudinal validation, broader participant inclusion, and more diverse task conditions.
Reviewer 2 Report
Comments and Suggestions for Authors
This study investigated mobility changes across the lifespan using smartphone accelerometry. The manuscript is detailed and well-written. However, there are several areas that should be addressed prior to acceptance. Specific comments follow.
Abstract
Unclear if adults in this study were all healthy, based purely on abstract.
Introduction
The introduction is well-written and thoroughly cites relevant literature. The rationale for the study is explained based on investigating the ecological validity of dual-tasking and the effect of external loading on mobility.
The hypotheses indicate that the investigators are just looking for changes in the outcome measures, but no a priori hypotheses are made regarding the direction of change for any of these variables? Also unclear why stair ascent and descent times were predicted to change in older adults only, when in the prior paragraph the authors indicate that stair negotiation may be affected by early aging.
Methods
Not clear how the authors arrived at an effect size of 0.4.
How did the investigators determine if the participants may have had “other comorbidities that could affect mobility”?
While the investigators have provided prior citations in the introduction regarding the impact of 10% body weight on mobility, this amount of weight does seem more than that required to induce mobility/balance changes, especially in older or less active adults. Perhaps indicate the activity level of participants to understand any effects here?
Confused by the calculation of muscle power, as it seems to redundantly include both the body mass and division by body weight (which is presumably body mass *g).
Line 188: Curious to know how many outliers were removed using this decision and if any analysis was performed on the individuals who demonstrated these outlier values.
Results
Results are presented thoroughly, though there does seem to be significant redundancies in the written text and figures/tables.
All figure captions do not require all acronym notes. E.g. for Figure 2, there doesn’t seem to be a DTW indicated, though it is in the caption.
Discussion
Lines 305 – 307: I am not convinced that changes in variability or similarity are indicative of “mobility deterioration”. As you have conceivably recruited healthy individuals, couldn’t these age-related changes instead be indicative of strategies utilized to ensure safe ambulation? This is also somewhat emphasized by the stride time variability results discussed later (lines 324-326). Though this is discussed well in the subsequent paragraphs, I’d be careful in how the phrase “mobility deterioration” is utilized.
Lines 372-373: 10% is too light? This again seems too counterintuitive, as participants are carrying ~7kg, which must be tiring when placed in a shopping bag. The older adults were also heavier and therefore carried a slightly heavier load, while also being potentially of lower strength?
Reviewer 3 Report
Comments and Suggestions for Authors
1. there number of references showing et al. Correct names should be added.
2. You mention twice (lines 371, 379) that the results are contrasting your hypothesis, but no hypothesis was stated in the body of the paper..
